# Infective Endocarditis among Pediatric Patients with Prosthetic Valves and Cardiac Devices: A Review and Update of Recent Emerging Diagnostic and Management Strategies

**DOI:** 10.3390/jcm12154941

**Published:** 2023-07-27

**Authors:** Mohamed Dardari, Eliza Cinteza, Corina Maria Vasile, Paul Padovani, Radu Vatasescu

**Affiliations:** 1Faculty of Medicine, Carol Davila University of Medicine and Pharmacy, 050474 Bucharest, Romania; dardarimohamed@yahoo.com (M.D.); radu_vatasescu@yahoo.com (R.V.); 2Electrophysiology and Cardiac Pacing Lab., Clinical Emergency Hospital, 014461 Bucharest, Romania; 3Interventional Cardiology Compartment, Marie Sklodowska Curie Children Emergency Hospital, 041451 Bucharest, Romania; 4Department of Pediatrics, Carol Davila University of Medicine and Pharmacy, 050474 Bucharest, Romania; 5Pediatric and Adult Congenital Cardiology Department, M3C National Reference Centre, Bordeaux University Hospital, 33600 Bordeaux, France; 6Nantes Université, CHU Nantes, Department of Pediatric Cardiology and Pediatric Cardiac Surgery, FHU PRECICARE, 44000 Nantes, France; padovanipaul17@gmail.com

**Keywords:** infective endocarditis, cardiac valves, cardiac device infections, diagnosis, imaging and microbiology

## Abstract

Infective endocarditis (IE) is a disease of the endocardium, which leads to the appearance of vegetation on the valves, cardiac structures, or, potentially, vascular endothelium of the heart. The risk of IE can be increased more than 140 times by congenital heart disease (50–59% of all IE), particularly if cyanotic. An increase in mortality may result from IE in patients with a complex cardiac pathology or patients with an implanted prosthetic material, most frequently conduits in a pulmonary position. Cardiac implantable electronic devices (CIED) infective endocarditis is a life-threatening complication representing 10% of all cases of endocarditis. Common signs of presentation are often fever and chills; redness and swelling at the pocket of the pacemaker, including the erosion and exteriorization of the device; and life-threatening sepsis. The use of intracardiac echocardiography for the diagnosis of IE is an innovative method. This may be needed, especially in older children undergoing complex cardiac surgery, when transthoracic echocardiography (TTE) and transesophageal echocardiography (TOE) failed to provide a reliable diagnosis. The 2018 European Heart Rhythm Association (EHRA) experts’ consensus statement on transvenous lead extraction recommends complete device removal and antimicrobial therapy for any device-related infection, including CIED-IE. The most detected microorganism was Staphylococcus Aureus. In addition, cardiac surgery and interventional cardiology associated with the placement of prostheses or conduits may increase the risk of IE up to 1.6% for Melody valve implantation. Our manuscript presents a comprehensive review of infective endocarditis associated with cardiac devices and prostheses in the pediatric population, including recent advances in diagnosis and management.

## 1. Introduction

Infective endocarditis (IE) is a disease of the endocardium, which leads to the appearance of vegetation on the valves, cardiac structures, or, potentially, vascular endothelium of the heart. This disease is estimated to affect 0.43–0.69 cases per 100,000 children [1,2]. A pre-existing heart condition or central venous catheter is more likely to cause it in males, usually schoolchildren and adolescents. IE can be increased more than 140 times by congenital heart disease (CHD) (50–70% of all IE), particularly if cyanotic, such as tetralogy of Fallot, pulmonary atresia with ventricular septal defect, transposition of great vessels or those that are not cyanotic, ventricular septal defect, pulmonary stenosis, aortic stenosis, and coarctation [3,4]. Other risk factors for infective endocarditis are presented in Figure 1. The most predisposing factors for adult patients are degenerative valve disease or surgical valve replacement with prosthesis implantation, which is significantly more common in adults than in children (Figure 1) [5,6,7].

Around 90% of children with CHD are expected to survive to adulthood. For some conditions, reintervention may be necessary after initial treatment. There may be a need for prosthetic materials, valve-related interventions, or intracardiac devices under these conditions, contributing to the high risk of IE in this group of patients. IE accounts for approximately 4% of all admissions in a tertiary care facility, and its associated mortality is about 7% [8,9]. An increase in mortality may result from IE in patients with complex cardiac pathology or patients with an implanted prosthetic material, most frequently conduits in a pulmonary position [10,11]. Mortality significantly increases when the disease is staphylococcal [8] or fungal (75–90%) or when the condition is complicated by heart failure or widespread dissemination—strokes or distal emboli. This can lead to an impressive increase in mortality, affecting up to 90% [12].

## 2. Diagnosis of IE in Patients with Devices or Prostheses

In infective endocarditis, the diagnosis is based on the modified Duke criteria, the current gold standard for diagnosing infective endocarditis, which can be followed in Figure 2 [13,14].

Only specific paraclinical investigations are considered in the major criteria—repeated positive blood cultures for specific germs; echocardiographic images indicating vegetation, an abscess, a perforation, aneurysms, pseudoaneurysms, fistulas, or prosthesis dehiscence; or more recent investigations showing abnormal activity at the implant site of a valve prosthesis, which is detectable by F-FDG PET/CT.

There are minor criteria as well, including predisposing factors, fever, and vascular manifestations (arterial emboli, septic pulmonary infarctions, mycotic aneurysms, intracranial or conjunctival hemorrhages, and Janeway lesions—Figure 3), as well as immunological manifestations (glomerulonephritis, Osler nodules, Roth spots, and a positive rheumatoid factor), positive blood cultures that do not meet the major criteria, or serological evidence of infection [1].

### 2.1. Microbiological Diagnosis

The cornerstone for microbiological diagnosis is positive blood cultures identifying the pathogen and the sensitivity to antimicrobial therapy. In order to determine the pathogen responsible for IE, it is imperative that blood samples are collected for microbiological examination as soon as possible from all patients suspected of having IE, especially if a patient is experiencing a fever of unexplained origin, a new heart murmur, a post-implantation device, a post-procedure valvuloplasty, or a previous history of endocarditis. The process involves taking three blood cultures through separate venous punctures on the first day of fever. If there is no rise after two days of incubation, 2–3 more blood samples should be taken [15]. These cultures can most frequently identify *Streptococcus viridans*, *Streptococcus bovis*, *Staphylococcus aureus*, and microorganisms from the HACEK group (*Haemophilus parainfluenzae*, *H. aphrophilus*, *H. paraphrophilus*, *Actinobacillus actinomycetemcomitans*, *Cardiobacterium hominis*, *Eikenella corrodens*, *Kingellas*, etc.). Due to children’s relatively minor circulating blood volume, the blood samples should be smaller. Infants and younger children should be limited to 1–3 mL, and older children should be limited to 5–7 mL. Bacteremia is usually constant in IE, so there is no need to expect pyrexia. Isolated growth in a single sample should be cautiously considered because of the high possibility of contamination [16]. It is important that blood cultures are completed without delay, especially during fever peaks and before initiating empiric antimicrobial therapy. Usually, a single positive blood culture is enough to establish the diagnosis of IE in the given clinical setting. When a pathogen is identified, immediate specific antimicrobial therapy is started, and the blood cultures should be repeated after 48–72 h to evaluate the effectiveness of treatment. Several improvements have been proposed to speed up the process of the detection and identification of pathogens. One of the most recent procedures for rapid bacterial identification is based on peptide spectra obtained by matrix-assisted laser desorption ionization time-of-flight mass spectrometry. This technique recently demonstrated its usefulness in clinical microbiology; it also has the potential for the direct identification of bacterial colonies in the blood culture bottle supernatant [17]. Staphylococcus aureus is the most common pathogen in both children and adults without any structural disease [18], while it is less frequent (30%) in the CHD group [19]. Streptococci infections, particularly those caused by the viridans group, are also widespread, especially in children with underlying heart disease, including CHD (43%) [19,20,21].

Even with advances in microbiological diagnostic techniques, such as polymerase chain reaction and serology, and the sensitivity of the modified Duke classification, approximately 12% of IE cannot be labeled as “definite” [22].

Approximately 2.5%–31% of blood cultures remain sterile, especially when antibiotic therapy is pre-administered, in the presence of slow-growing bacteria, including anaerobes, the HACEK group, *Abiotrophias*, *Brucellas*, *Bartonellas*, *Legionellas*, and *Mycoplasmas* or when intracellular microorganisms such as *Coxiella burnetii* are the cause. Several types of bacteria require specific culture media and conditions, such as buffered charcoal yeast extract (BCYE) agar for *Legionella* spp. and L-cysteine-enriched medium for *Abiotrophia* spp. If fastidious microorganisms or unusual pathogens are suspected, the microbiology laboratory should be strengthened so that more prolonged incubation and more specific tests can be used. Additionally, up to six weeks of incubation may be necessary for slow-growing bacteria in some cases [21]. However, after cardiac surgery, *S. aureus* and coagulase-negative staphylococci are more common [23,24,25].

### 2.2. Imagistic Techniques

Usually, transthoracic echocardiography (TTE) can provide sufficient information to recognize the changes and for follow-up. The limitations of TTE can arrive in complex congenital heart disease, especially in patients after cardiac surgery. Moreover, shunts and conduits are challenging to distinguish from other prominent cardiac structures and surgical remnants. This is because it is challenging to distinguish vegetation from such structures. It is estimated that false-negative results can reach 70% for TTE after cardiac surgery with complex CHD [26]. Children with a weight of fewer than 60 kg have comparable diagnostic performance to adults with transesophageal echocardiography (TEE). This is despite having a higher resolution and, therefore, increased sensitivity. The sensitivity of TTE in pediatric patients proved to be 97%. But, if concluding that images cannot be obtained, the following step is using transesophageal echocardiography [13,27].

Children with congenital heart disease can benefit from transesophageal echocardiography, especially those who have already undergone surgery or intend to undergo surgery for IE in the near future. It is usually necessary to monitor intraoperative cardiac surgery using transesophageal echocardiography. Despite its advantages, transthoracic echocardiography has decreased diagnostic performance in children with structurally normal hearts and negative microbiological tests when the probability of IE is low. A TEE should be conducted in patients with clinical suspicion of IE and a non-diagnostic or negative TTE, a prosthetic valve, or an intracardiac device. It is necessary to repeat this procedure in 5–7 days if there is a high clinical suspicion and a negative initial TEE examination [13].

Therefore, a few specific points need to be considered prior to a transesophageal ultrasound in children. Sedation or general anesthesia may be necessary, and a probe appropriate to the child’s weight should be used [28]. When vegetation is embolized or is smaller than 2 mm, or the patient lacks vegetation, even TEE examinations can be inconclusive [29].

Intracardiac echocardiography is a particularly valuable tool in the diagnosis of IE. As previously mentioned, conventional transthoracic and transesophageal echocardiography may have poor diagnostic performance in children with cardiac devices or previous corrective surgery. Intracardiac echocardiography, a very flexible technique, can overcome these limitations [30,31,32].

Recent years have witnessed an increase in the use of advanced cardiovascular imaging modalities such as magnetic resonance imaging, computed tomography, and single photon emission CT (SPECT)/CT and positron emission computer tomography (PET-CT) with the administration of ^18^F-FDG. Patients with suspected IE and diagnostic difficulties may benefit the most from these new imaging techniques, which can also be used to monitor response to therapy (i.e., ^18^F-FDG PET/CT).

SPECT/CT imaging relies on the use of autologous radiolabeled leucocytes (^111^In-oxine or ^99m^Tc-hexamethyl propylene amine oxime) that accumulate in a time-dependent fashion in late images versus earlier images [33], whereas PET/CT is generally performed using a single acquisition time point (generally at 1 h) after the administration of ^18^F-FDG, which is actively incorporated in vivo by the activated leucocytes, monocyte macrophages, and CD4^+^ T-lymphocytes accumulating at the sites of infection. The main added value of these new imaging techniques is reducing the rate of misdiagnosing IE in patients labeled as possible IE by the Duke criteria and for detecting embolic and metastatic infectious events [34].

There are several limitations to the use of ^18^F-FDG PET/CT. Recent cardiac surgery may still provoke a local inflammatory pattern, which results in a non-specific ^18^F-FDG uptake, while other conditions may mimic infection such as active thrombi, soft atherosclerotic plaques, vasculitis, primary cardiac tumors or cardiac metastasis from a non-cardiac tumor, and foreign body reactions [35].

Radiolabeled WBC SPECT/CT is more specific for the detection of IE and infectious foci than ^18^F-FDG PET/CT and should be preferred in all situations that require enhanced specificity [36]. The disadvantages of scintigraphy with radiolabeled WBC are the requirement of blood handling for radiopharmaceutical preparation; the duration of the procedure, which is more time-consuming than PET/CT; and a slightly lower spatial resolution and photon detection efficiency compared with those of PET/CT. Multislice CT is superior to TEE in evaluating IE-related valve abnormalities, such as abscesses or pseudoaneurysms [37].

## 3. Cardiac Devices in Children: CIED

### 3.1. Overview

The indications for cardiac implantable electronic device (CIED) implantation are widening in adults and the pediatric population. Pacing for congenital heart block is now more common, as indications now include less severe criteria for pacing, such as a mean heart rate of <50, prolongation of QTc, broad QRS escape rhythm, complex ventricular arrhythmias detected on resting ECG or Holter monitoring, syncope, and symptoms during effort. The currently estimated number of active devices has reached 4.5 million, with over 1 million new implants yearly [38]. In the pediatric population referred to cardiac surgery, heart block and permanent pacemaker/defibrillator implantation incidence is 1% [39]. While pacing and defibrillator leads are most frequently epicardially implanted in children, most of them, at some point, require an upgrade to dual-chamber endocardial transvenous pacing during their adult life.

### 3.2. Infective Endocarditis in Patients with CIED

CIED infective endocarditis is a life-threatening complication representing 10% of all cases of endocarditis [40]. Infection may primarily involve the pocket after direct manipulation (e.g., changing the generator). It may disseminate to the leads, producing multiple vegetations, or directly originate from the leads during bacteremia, secondary to a minor infection outbreak. In addition to the typical risk factors for IE (renal failure, corticosteroid use, congestive HF, and diabetes mellitus), other factors related to the surgical procedure may play a role in IE-CIED (e.g., the type of procedure, device revision, the use of temporary pacing, the use of antimicrobial prophylaxis, and the use of anticoagulation) [41]. Common signs of presentation are often fever and chills; redness and swelling at the pocket of the pacemaker, including erosion and exteriorization of the device (Figure 4 and Figure 5); and life-threatening sepsis [42]. The modified Duke criteria are used for diagnosis, and the management consists of individualized antimicrobial therapy and complete device removal, which is known to be the only safe and effective treatment in the long term [43]. According to the Euro-Endo registry, Staphylococci account for 60–80% of cases, with a significant proportion of S. aureus (approximately 50%) [27].

### 3.3. Diagnosis

The diagnosis of CIED-IE is complex and requires additional time; one cohort study reported 48 days [44]. The 2020 EHRA international consensus document on preventing, diagnosing, and treating cardiac implantable electronic device infections mentions different types of infections [45]. Infection may present as a superficial incisional infection (not involving the generator pocket), which usually resolves with only medical treatment and superficial wound care, pocket infection, and systemic and infective endocarditis infection. The latter three mandate the complete removal of all material. The diagnosis of CIED-IE is complex and may require a long time to diagnose and treat. Infective endocarditis is commonly clinically suspected and is usually confirmed using multiple diagnostic tests, including echocardiography, computer tomography, and blood cultures, based on the Duke criteria to determine the probability of diagnosis as (a) rejected, (b) possible, or (c) definitive. Fluorodeoxyglucose (^18^F-FDG) PET/CT has found its way to improve the diagnostic accuracy of the modified Duke criteria in patients with suspected infective endocarditis with prosthetic valves or cardiac devices [46]. Reports suggested that intracardiac lead vegetations were significantly more frequently found in patients with heart and renal failure [47], while older patients, diabetic patients, and patients not receiving anticoagulation therapy had independently associated with more extensive vegetation [30]. Larger vegetation mandated the need for open surgical interventions, which in turn caused significant complications [48]. The overall mortality rate in the literature ranged from 4% to 36%, with a 10-year mortality rate reported between 53.3% and 80%.

### 3.4. Management

The 2018 EHRA expert consensus statement on transvenous lead extractions and the 2020 EHRA international consensus document on how to prevent, diagnose, and treat cardiac implantable electronic device infections recommend complete device removal and antimicrobial therapy for any device-related infection, including CIED-IE [44,49]. The most detected microorganism was Staphylococcus Aureus.

The most common indication for lead extraction is device-related infection. In one study describing the transvenous removal of 135 leads, 77.8% of extracted leads were for pocket infection, followed by 9.8% for infective endocarditis [50]. Transvenous lead extraction is now a well-defined procedure frequently used for devices and lead extraction for various indications. The technique consists of ligation and continuous gentle traction of the extravascular end of the targeted lead to maintain constant tension using a surgical suture thread. After which, the Byrd dilator sheath (Figure 6) is used for indwelling intravascular leads by continuously rotating and advancing the sheath around the targeted indwelled lead along its length and down to its tip. At the same time, it is kept under tension, the tissue surrounding the cardiac leads ruptures, and the lead becomes free from intravascular and intracardiac adherences. The technique was proved to be highly successful (>90% in most papers), with a low complication rate [51,52]. When TLE fails, open chest surgical removal is indicated.

## 4. Cardiac Surgery or Interventional Procedures in Children

Cases of IE after cardiac surgery or interventional procedures in children are sporadic. It was observed that IE incidence has increased with the implantation of pacemakers, replacement of intracardiac prosthetic valves, and improved endovascular treatment of patients with IE associated with CHD in recent years [50]. In addition to being susceptible to bacterial colonization, prosthetic materials are also difficult to sterilize [50]. Infective endocarditis may occur in patients requiring cardiac device implantation during initial hospitalization or after discharge from the hospital. Staphylococcus aureus is the most prevalent pathogen among healthcare-associated microorganisms, accounting for approximately 30% of cases [40]. It is essential to highlight that in serial blood cultures from patients who develop bacteremia, Staphylococcus aureus was reported to be found in 70% of those examined.

### 4.1. Cardiac Surgery and Prosthetic Valves, Conduits, and Other Material Implants

IE in children is rarely studied, and very few data are available from Level III hospitals, including cardiac surgery centers. Based on these data, many IE cases are related to surgery. In a study conducted in Turkey on 45 pediatric patients with IE, 30 had a history of congenital heart disease. At the time of IE, 19 patients had foreign material inside their hearts (biological valved conduit in the pulmonary position, mitral prosthetic valve, aortic prosthetic valve, PTFE graft, or dual-chamber pacemaker lead). After the prosthetic valve was replaced, three months, two years, and eight years passed before the diagnosis of IE was made. As a result of reviewing the whole group of patients following cardiac surgery, it was found that 10% developed IE within six months [10]. A further interesting observation for this group of patients was that one-fourth (25.5%) had pneumonia, acute gastroenteritis, an open wound infection, or untreated dental caries one month before cardiac surgery [10]. Patients under 3 years old who have undergone cardiac surgery to correct congenital heart disease have a five-fold increased risk of developing infective endocarditis within 6 months [53,54].

In developed countries, IE related to right heart pathology has decreased due to the provision of appropriate cardiac care and cardiac surgery. In contrast, IE related to prosthetic valves affecting the left heart has increased [10,54,55,56,57,58,59].

IE on the non-congenital affected pulmonary valve is rare compared to congenital pulmonary pathologies. Replacing the affected valve or conduit with a bioprosthesis or an allograft/xenograft worked well [60].

Aortic valve repair in children is desirable but only sometimes feasible. An aortic valve replacement in children may involve the use of a mechanical prosthesis or the Ross procedure (replacing the aortic valve with the pulmonary valve and placing a conduit in the pulmonary position). In adults, replacement, either surgical or interventional, is associated with several risk factors for IE. These include a younger age, the male gender, a higher body mass index, and an elevated post-deployment gradient [61].

A mitral valve repair may not always be possible, especially in children. It may be necessary for a mitral valve replacement to be performed in children with congenital mitral stenosis, mitral insufficiency, or a repaired atrioventricular septal defect and patients infected with mitral valve endocarditis. Replacement of the mitral valve with a mechanical prosthesis may increase the risk for IE. A 5-year survival rate of 96% and a 10-year survival rate of 80% were expected for patients who avoided postoperative endocarditis [62].

### 4.2. Transcatheter Valve Implantation and Other Interventional Procedures

Reconstruction of the right ventricle ejection tract is a standard procedure, especially in adult congenital heart disease, but it is also becoming more common in children (Figure 7). Depending on the severity of the condition, patients may require multiple surgeries in the future [63]. To avoid or reduce multiple reinterventions, Phillip Bonhoeffer initially described the Medtronic Melody valve implantation procedure on humans in 2000 [64]. The most crucial disadvantage of this procedure is the increased risk of infective endocarditis compared to surgery (almost three times higher for the Melody valve—1.6%), especially in the first two years after implantation [65,66]. Other models of valves with advantages and disadvantages appeared afterward, including the Edwards Sapien valve, which proved to have a reduced incidence of IE after implantation [67]. New self-expandable valves are used in the pulmonary position (Harmony, Venus *p*-valve, and others), and new results are expected. Studies also indicated that the risk of IE is more directly related to the valve tissue (e.g., bovine jugular veins versus others) than the valve implantation method. Contegra conduits and Melody valves derived from the same biological substrate demonstrate a significantly higher risk of IE than other biological pulmonary valve substrates (e.g., homografts, Sapien valves, and Hancock valves) [63,68]. Usually, prolonged antibiotic treatment for at least 6 weeks is enough to treat IE, but, in some cases, extraction of the prosthesis and pulmonary surgical replacement may be necessary.

When comparing surgical and transcatheter replacements of the aortic valve over 10 years in England in adults, the surgical cohort had a higher IE incidence rate of 4.8%, compared to 3.6% in the transcatheter cohort using the Edwards Sapien valve [61]. Figure 7 depicts the case of a 5-year-old boy who underwent surgery to address severe aortic insufficiency caused by a bicuspid valve. The surgical procedure utilized the Ross technique, inserting an 18 mm Contegra tube in the pulmonary tract.

Few reported cases of IE on defect closure devices involve atrial or ventricular septal defects. In the pediatric population, some patients have occurrences at long intervals after the defect has been closed, such as after seven months, two years, or six years. However, IE was also observed after two months of ventricular septal defect interventional closure. The prosthesis’ lack of endothelization causes these complications. The Septal Occluder and Figulla Flex II ASD Occlutech prostheses were used. The device was removed, the defect was surgically repaired, and no other complications occurred. In addition, patients received prolonged antibiotic treatment (Figure 8) [69,70,71,72,73].

## 5. Updates on Guidelines, Antibiotic Resistance and Regimens, and Prophylaxis

Antimicrobial therapy is required to manage IE, focusing on antibiotics for a prolonged period. The choice of antibiotic and the duration of the administration is based on the causative pathogen, potential antibiotic resistance, and type of infected material, whether native tissue, prosthetic valve, or cardiac device [74]. The antimicrobial treatment of IE is extensively described in the European Society of Cardiology (ESC), American Heart Association (AHA), and Stichting Werkgroep Antibioticabeleid (SWAB) guidelines [9,75].

The pathogens involved range depending on the underlying conditions. In addition, there have been several changes in the epidemiology of IE in recent decades. Worldwide, *S. aureus* remains the dominant causative pathogen in children and adults [20]. Infections with streptococci, mainly from the viridans group, are equally widespread [20]. Particularly in children, Staphylococcus species are the most common (notably in children without cardiac disease), followed by Streptococcus species [19,31]. Nevertheless, in children with underlying cardiac disease, *Streptococcus viridans* is the most common cause [18,19,32]. With the same pattern as in adults, IE from Gram-negative microorganisms are rare in children [19,53,76].

Evidence for antibiotic prophylaxis to prevent IE is based on observational data suggesting that invasive dental procedures or oral procedures may increase the risk of IE in patients at high risk of IE. Antibiotic prophylaxis may decrease this risk. However, the benefit of antibiotic prevention has yet to be definitively established, even in extensive clinical studies [76,77,78].

By 2021, the AHA published a statement reviewing the impact of the 2007 guidelines on the incidence and outcomes of IE due to viridans group streptococci, with additional recommendations on the use of antibiotic prophylaxis [79]. The 2015 ESC guidelines align with previous US guidelines [13].

The frequency of IE limits the benefits of antibiotic prophylaxis without an identifiable prior procedure, the low rate of IE following invasive dental procedures without precaution, and the failure potential of prevention. Thus, it was estimated that even if antibiotic prevention is fully adequate, less than 10% of all endocarditis cases could be prevented by using antibiotic therapy before procedures [78]. In addition, the observed rates of adherence to antibiotic prophylaxis guidelines for patients at high risk of IE are low (ranging from 26% to 50%) [79,80].

The antibiotic therapy depends on the type of microorganism, the susceptibility of the antibiogram, and the risk factors of the patient, including underlying cardiac disease. Regarding the duration of antibiotic treatment, it should generally be extended in patients with prosthetic valves or corrective surgery with the implantation of prosthetic materials [13,17]. Intravenous antibiotic treatment should be used in children, avoiding intramuscular administration due to the risk of complications and lower muscle mass. Antibiotics with bactericidal action are preferred to bacteriostatic ones.

Outpatient treatment could be considered in children without complications or high-risk factors; in children with negative control blood cultures, who remain afebrile; and in children whose therapeutic adherence, parental cooperation, and close supervision by parents and a home nurse can be guaranteed [81,82,83].

Empiric antibiotic therapy should be started as soon as possible in critically ill patients without an available antibiogram or in cases with negative blood cultures.

When starting treatment for a patient with prosthetic valve endocarditis, it is necessary to identify the causative microorganism, select the appropriate antibiotic, assess the possible complications of endocarditis, and evaluate the need for surgery when antibiotic treatment fails if the infection causes a valve abscess.

Empiric antibiotic treatment should be started after obtaining three blood cultures. This treatment should aim to cover Gram-positive and Gram-negative bacteria. This empirical treatment should include vancomycin, gentamicin, and carbapenem or anti-pseudomonal cefepime. When the causative microorganism is identified, treatment should be tailored to cover this causative agent. The American Heart Association recommends at least six weeks of antibiotic treatment [13,79].

## 6. Conclusions

Through this overview, we highlight the importance of prevention in patients at high risk of infective endocarditis, which can often be life-threatening, and the need to develop comprehensive guidelines for these patients. The modern management of IE and CIED-IE prostheses, conduits, or defect closure devices requires a proper and collaborative integration of diagnostic tools and therapeutic strategies. Multimodal imaging is a crucial part of diagnosis, where different techniques provide additional and unique information. Clinicians and imaging specialists should be aware of the strengths and limitations of each approach to appropriately interact with other specialists and, therefore, optimize patient management and improve outcomes.

## Figures and Tables

**Figure 1 jcm-12-04941-f001:**
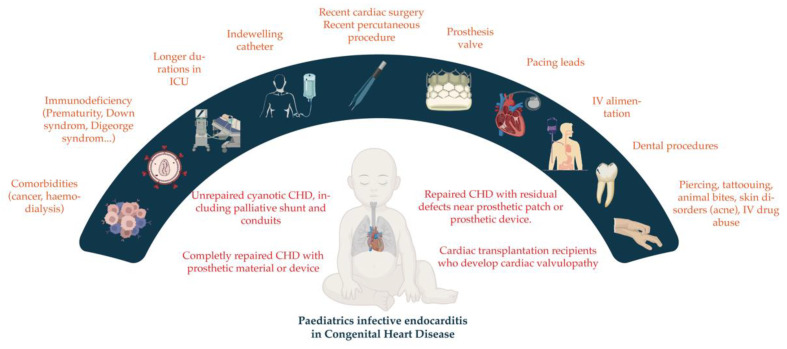
Patients with predisposing cardiac conditions at high risk for IE Prophylaxis.

**Figure 2 jcm-12-04941-f002:**
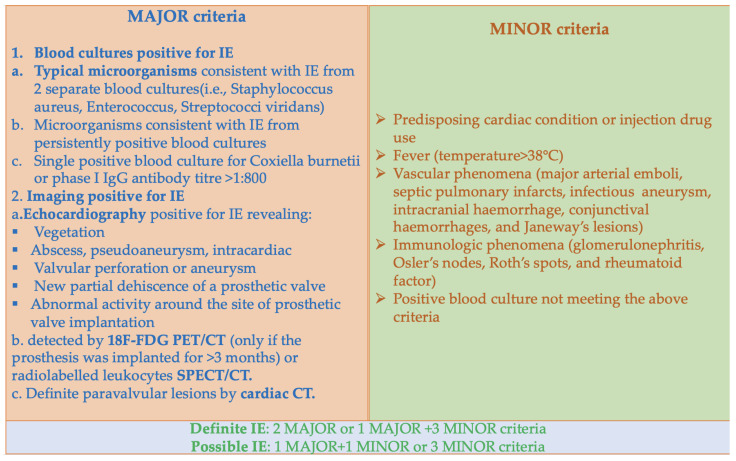
Modified Duke criteria for the diagnosis of infective endocarditis [13,14].

**Figure 3 jcm-12-04941-f003:**
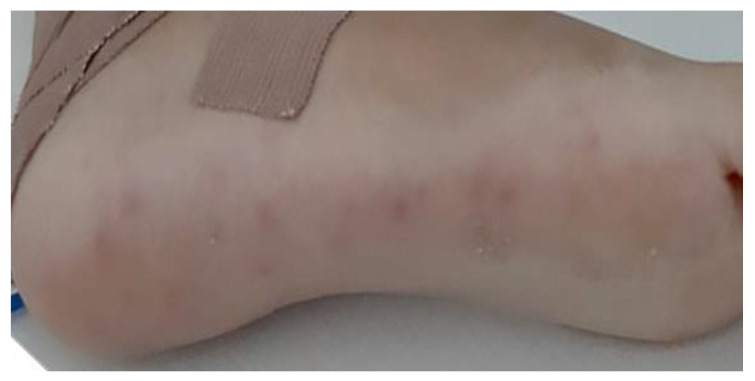
Janeway lesions in a 7-year-old boy with fungal endocarditis.

**Figure 4 jcm-12-04941-f004:**
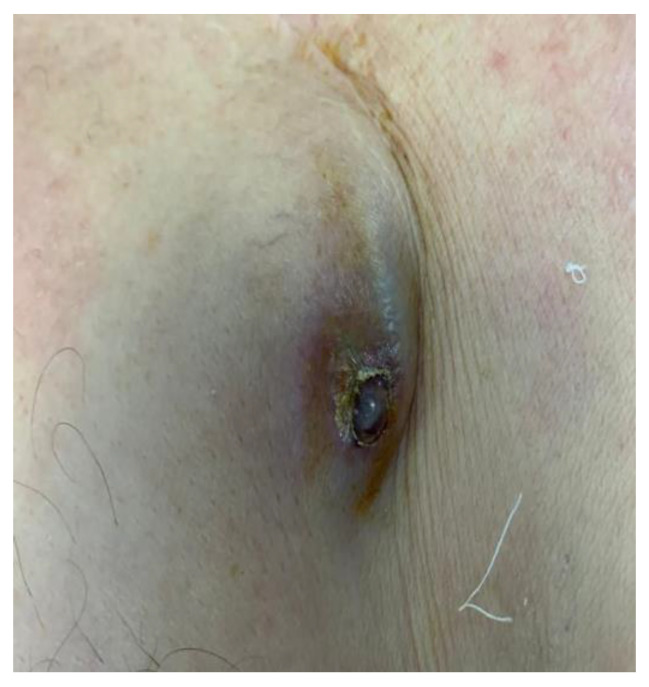
Infected pacemaker presenting as skin erosion with exteriorization of the adherent-to-skin device.

**Figure 5 jcm-12-04941-f005:**
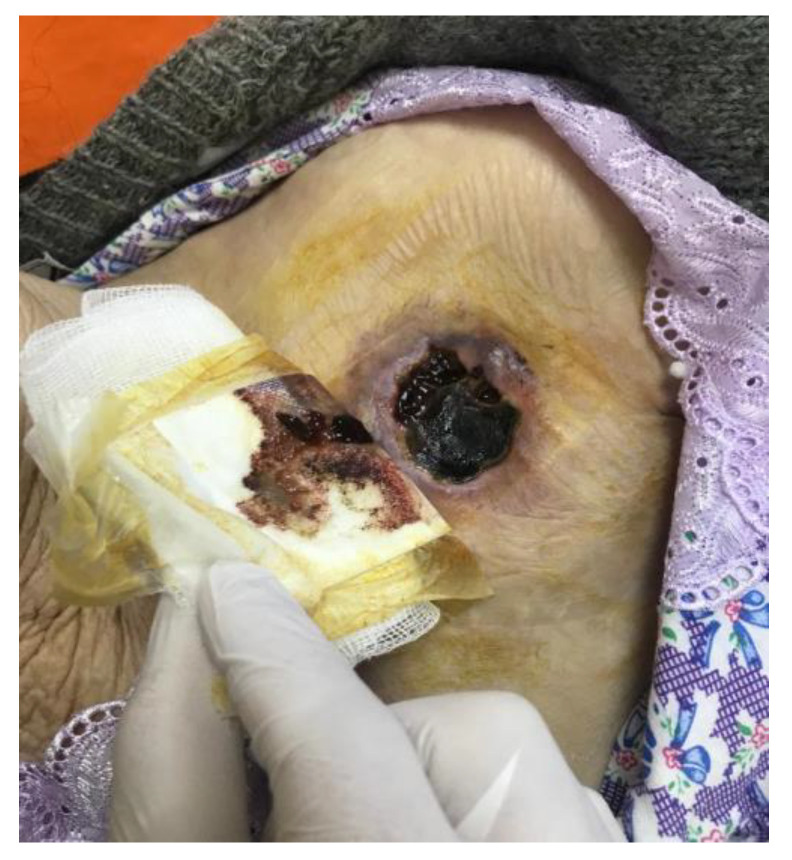
Skin necrosis of infected pacemaker pocket. Image courtesy of Iorgulescu Corneliu, Clinical Emergency Hospital of Bucharest, Romania.

**Figure 6 jcm-12-04941-f006:**
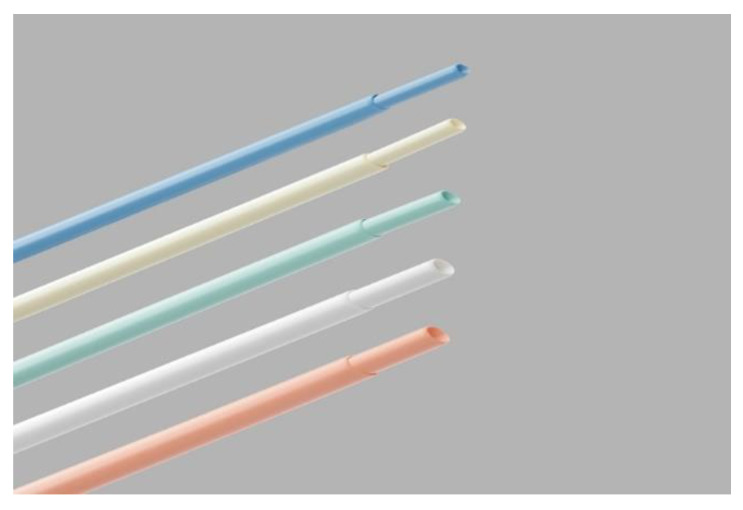
Byrd Dilator Sheath Telescoping Polypropylene used for traction and indwelling leads through their lumen by continuous mechanical rotation and advancement along each lead for rupturing intravascular adherences to free the lead for extraction. Several dimensions are available (7 to 12 French). Upscaling in the same procedure is frequent.

**Figure 7 jcm-12-04941-f007:**
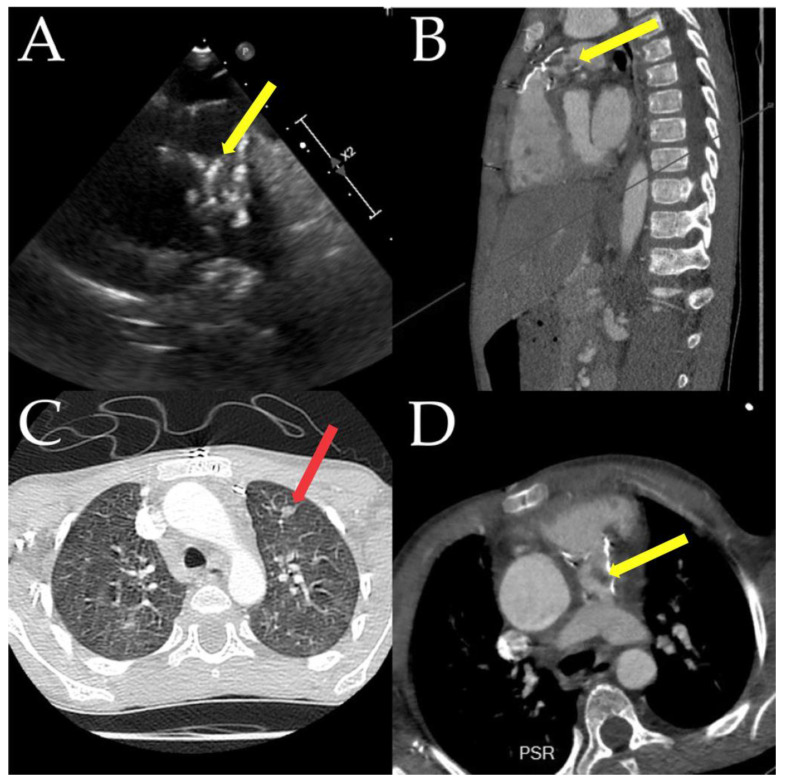
A 5-year-old boy was operated on for a severe aortic insufficiency (bicuspid valve) by the Ross technique with an 18 mm Contegra tube insertion in the pulmonary tract. An increase in gradients in the RV-PA conduit (**A**) was demonstrated during follow-up. Further tests (**B**,**D**) revealed methicillin-sensitive Staphylococcus aureus endocarditis ((**A**,**B**,**D**), yellow arrow) complicated by septic pulmonary emboli ((**C**), red arrow). The boy needed a new surgery after 6 weeks of antibiotics. Image courtesy of Benbrik.

**Figure 8 jcm-12-04941-f008:**
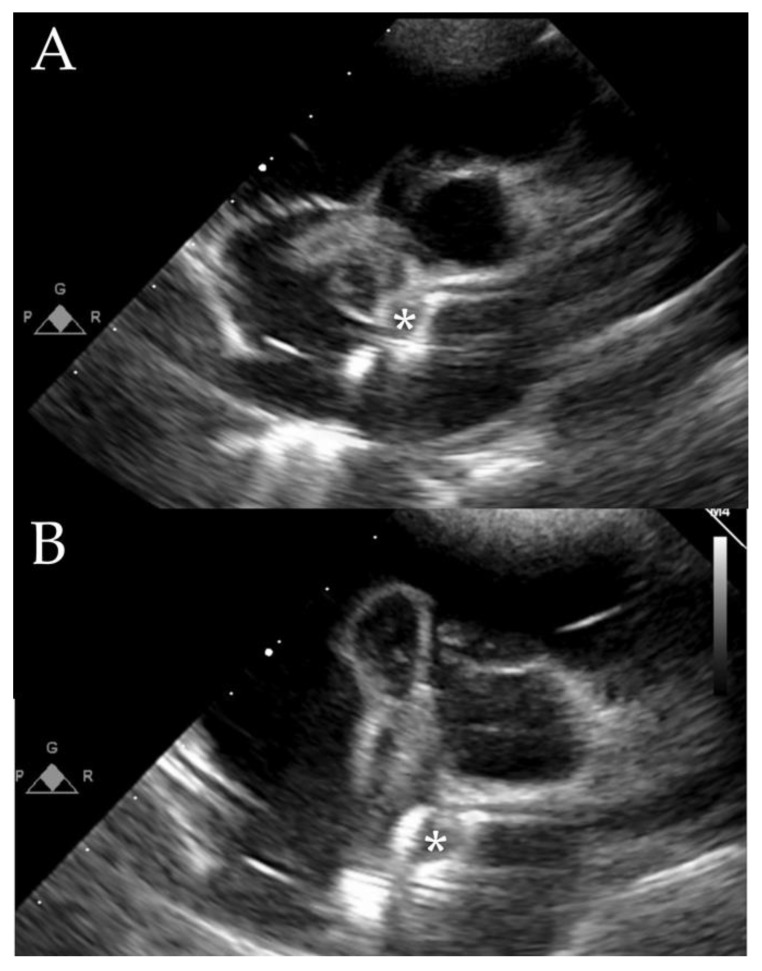
Methicillin-resistant Staphylococcus aureus endocarditis was discovered one year after percutaneous ASD closure (*: ASD closure device). (**A**): shows the vegetation attached to the ASD closure when the tricuspid vakve is closed. (**B**):shows the movements of this valve extending down the tricuspid valve and creating a blood flow obstacle during the right ventricular filling. The primary route of infection was an impetiginized skin lesion on the forehead. After identification, empiric antibiotic therapy with Cefotaxime + Gentamicin was switched to Vancomycin, Ceftazidime, and Gentamicin. Surgery was one week after the start of antibiotic therapy due to the size of the vegetation (3 × 2 cm), which did not decrease under effective antibiotic therapy. Image courtesy of Le Seven.

## Data Availability

Data are available on request from the corresponding authors.

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
