# Peer review of "Infective Endocarditis among Pediatric Patients with Prosthetic Valves and Cardiac Devices: A Review and Update of Recent Emerging Diagnostic and Management Strategies"

_jcm, 2023, doi:10.3390/jcm12154941_

Round 1
Reviewer 1 Report
This review provides an overview of infective endocarditis (IE) in pediatric patients with cardiac devices and prostheses. Congenital heart disease (CHD), especially cyanogenic CHD, poses a significant risk for IE. Complex cardiac pathology and prosthetic materials increase mortality risks. Cardiac implantable electronic devices (CIED) account for 10% of all endocarditis cases and are life-threatening. Common signs of IE include fever, chills, redness, and swelling at the pacemaker pocket. Microbiological examination combined with Imagistic diagnosis, such as transthoracic echocardiography (TTE), and transoesophageal echocardiography (TEE) and intracardiac echocardiography are used for IE diagnosis. Cardiac surgery and interventional cardiology procedures involving prostheses or conduits increase the risk of IE. Complete device removal and antimicrobial therapy are crucial for treating device-related infections. Staphylococcus aureus is the most frequently detected organism in IE cases. Overall, this manuscript provides a clear, comprehensive review of infective endocarditis associated with cardiac devices and prostheses in the pediatric population, addressing recent advances in diagnosis and management.
In addition, I have the following questions and suggestions:
1.I couldn't find Table 1 mentioned in the introduction section. Is it included in the supplementary data?
2.Although the authors have displayed Figures 4, 7, and 8, they did not reference these figures in the manuscript.
3.To enhance reader understanding, it would be helpful if the authors could highlight the targets in the panels of Figure 7.
4.It would be more illustrative if the authors could include examples of infectious endocarditis diagnosed by transthoracic echocardiography (TTE), transesophageal echocardiography (TEE), and intracardiac echocardiography.
The manuscript is well-organized and generally written in English. However, some sentences could benefit from moderate editing to enhance the overall clarity and flow of the manuscript. With a few improvements, the manuscript will become easier to read and follow. Overall, the quality of the English language is good, and with moderate revisions, it can be further improved.
Author Response
This review provides an overview of infective endocarditis (IE) in pediatric patients with cardiac devices and prostheses. Congenital heart disease (CHD), especially cyanogenic CHD, poses a significant risk for IE. Complex cardiac pathology and prosthetic materials increase mortality risks. Cardiac implantable electronic devices (CIED) account for 10% of all endocarditis cases and are life-threatening. Common signs of IE include fever, chills, redness, and swelling at the pacemaker pocket. Microbiological examination combined with Imagistic diagnosis, such as transthoracic echocardiography (TTE), and transoesophageal echocardiography (TEE) and intracardiac echocardiography are used for IE diagnosis. Cardiac surgery and interventional cardiology procedures involving prostheses or conduits increase the risk of IE. Complete device removal and antimicrobial therapy are crucial for treating device-related infections. Staphylococcus aureus is the most frequently detected organism in IE cases. Overall, this manuscript provides a clear, comprehensive review of infective endocarditis associated with cardiac devices and prostheses in the pediatric population, addressing recent advances in diagnosis and management.
We appreciate the recommendations and kind words of the reviewer.
In addition, I have the following questions and suggestions:
1.I couldn't find Table 1 mentioned in the introduction section. Is it included in the supplementary data?
We apologize for the mistake, it’s actually, Figure 1. We have revised it accordingly.
2.Although the authors have displayed Figures 4, 7, and 8, they did not reference these figures in the manuscript.
We have referred to all our figures in the manuscript.
3.To enhance reader understanding, it would be helpful if the authors could highlight the targets in the panels of Figure 7.
We have revised figure 7.
4.It would be more illustrative if the authors could include examples of infectious endocarditis diagnosed by transthoracic echocardiography (TTE), transesophageal echocardiography (TEE), and intracardiac echocardiography.
Unfortunately, we do not have any of those images available in our achieve.
Comments on the Quality of English Language
The manuscript is well-organized and generally written in English. However, some sentences could benefit from moderate editing to enhance the overall clarity and flow of the manuscript. With a few improvements, the manuscript will become easier to read and follow. Overall, the quality of the English language is good, and with moderate revisions, it can be further improved.
An English revision by a native speaker has been performed.
Reviewer 2 Report
Infective endocarditis among pediatric patients with prosthetic valves and cardiac devices: A review and update of recent emerging diagnostic and management strategies
In this review the authors summarize the current knowledge concerning infective endocarditis in children with prosthetic valves and cardiac devices. The review’s subject is quite interesting and offers important information regarding the diagnostic and treatment options in this subgroup of pediatric patients. However, there are several points that should be addressed by the authors.
Major comments
Main text
- Page 3, line 73-74: “In infective endocarditis, the diagnosis is based on the modified Duke criteria, the current gold standard for diagnosing infective endocarditis”.
According to the 2015 ESC guidelines for IE: “This classification has a sensitivity of approximately 80% overall when the criteria are evaluated at the end of patient follow-up in epidemiological studies. However, the modified Duke criteria show a lower diagnostic accuracy for early diagnosis in clinical practice, especially in the case of prosthetic valve endocarditis and pacemaker or defibrillator lead IE, for which echocardiography is normal or inconclusive in up to 30% of cases. Given the recent published data, the Task Force proposes the addition of three further points in the diagnostic criteria (Table 14): (1) The identification of paravalvular lesions by cardiac CT should be considered a major criterion. (2) In the setting of the suspicion of endocarditis on a prosthetic valve, abnormal activity around the site of implantation detected by 18 F-FDG PET/CT (only if the prosthesis was implanted for >3 months) or radiolabelled leucocyte SPECT/CT should be considered a major criterion. (3) The identification of recent embolic events or infectious aneurysms by imaging only (silent events) should be considered a minor criterion.” I suggest that this section and figure 2 should be revised accordingly, because it seems inaccurate.
- Page 5, line 146: “Although controversial, transesophageal echocardiography is valuable in children with congenital heart disease”. Why is it controversial? You mention above that in 70% of cases with IE in patients with CHD and cardiac surgery, TTE results are false-negative. I think that the whole section from line 135-152 should be rephrased, because it has poor English, and it is kind of confusing.
- Page 6: I think that the section concerning imaging diagnosis is incomplete. The role of PET/CT in prosthetic valve endocarditis (PVE) should be explained in more details. According to the 2015 ESC guidelines for IE: “Recently, nuclear techniques, particularly 18F-FDG PET/CT, have been shown to be useful for the diagnosis of PVE. The addition of abnormal FDG uptake as a novel major criterion for PVE has thus been pointed out. An algorithm for evaluation of patients with suspected PVE, including echocardiography and PET/CT has been suggested.” (Saby L, Laas O, Habib G, Cammilleri S, Mancini J, Tessonnier L, Casalta JP, Gouriet F, Riberi A, Avierinos JF, Collart F, Mundler O, Raoult D, Thuny F. Positron emission tomography/computed tomography for diagnosis of prosthetic valve endocarditis: increased valvular 18F-fluorodeoxyglucose uptake as a novel major criterion. J Am Coll Cardiol 2013;61:2374 –2382)
Fluoroscopy is also not mentioned as a way to diagnose IE in children with mechanical valves.
- Page 7, line 206: mention the novel 2019 International CIED Infection Criteria (EHRA international consensus document on how to prevent, diagnose, and treat cardiac implantable electronic device infections, Europace (2020) 22, 515–516, doi:10.1093/europace/euz246).
- Page 7, line 226-242: I suggest citing the most recent and relevant EHRA consensus document (2020) on CIED infections (EHRA international consensus document on how to prevent, diagnose, and treat cardiac implantable electronic device infections, Europace (2020) 22, 515–516, doi:10.1093/europace/euz246). You should add the recommendation for device and lead removal according to Table 8. You should also mention the duration of antibiotic treatment after the device removal (Table 9).
- Page 7, line 226-242: What is the management in case of CIED-IE of epicardial leads? It should be added.
- Page 9, line 281: “Aortic valve repair in children is desirable but only sometimes feasible. Replacement of the aortic valve in children is done using a mechanical prosthesis.” You should also add Ross procedure.
Minor comments
Abstract:
- “IE can be increased more than 140 times by congenital heart disease”. I think that it should be rephrased as follows: “The risk of IE can be increased more than…disease”.
- I suggest that the term “cyanotic”, instead of “cyanogenic” is used.
- “The most detected organism ... Aureus”. Perhaps instead of organism, you should use the word microorganism.
- “In addition, cardiac surgery and interventional cardiology associated with the placement of prostheses or conduits may increase the risk of IE up to 1.6% for Melody valve implantation.”. I think that you should incorporate this sentence with what you write in line 21-23. At the present position is kind of irrelevant.
Introduction:
- “Transposition of large vessels”. I suggest the word “great” instead of “large”.
- “Other risk factors for infective endocarditis may be followed in Table1”. I think what you meant is: “Other risk factors for infective endocarditis are presented in Table 1.”
Main text
- Page 3, line 74 “can be followed in Figure 2”. Again, I suggest that you use the phase is depicted or is presented instead.
- Figure 2: correct aureu, reveling
- Page 5, line 134: I think you meant “Imaging techniques”, instead of “imagistic diagnosis”
- Page 6, line 179: according to 2021 ESC guidelines for pacing the threshold for pacing in patients with congenital heart block is <50bpm.
- Page 6, line 188: I suggest that you should replace the term “handling” with “manipulation”.
- Page 8, line 272: “IE can develop within six months of undergoing cardiothoracic surgery in 272 children with CHD under three years, a risk that increases by five times in those children”: The sentence structure is confusing, I do not understand the meaning.
- Page 10, Figure 7: Please explain the abbreviations in the figure (VD-AP).
- Page 11, Figure 8: Instead of “probabilistic”, use the term “empirical”.
Conclusion
- Merge to one paragraph, as follows: “Through this overview, we would like to highlight the importance of prevention in patients at high risk of infective endocarditis, which can often be life-threatening, and the need to develop comprehensive guidelines for these patients. An up-to-date and modern management of IE and CIED-IE prostheses, conduits, or defect closure devices requires a proper and collaborative integration of diagnostic tools and therapeutic strategies. Multimodal imaging is a crucial part of diagnosis, where different techniques provide additional and unique information. Clinicians and imaging specialists should be aware of the strengths and limitations of each approach to appropriately interact with other specialists and therefore optimize patient management and improve outcomes.
Extensive editing of English language required.
Author Response
In this review the authors summarize the current knowledge concerning infective endocarditis in children with prosthetic valves and cardiac devices. The review’s subject is quite interesting and offers important information regarding the diagnostic and treatment options in this subgroup of pediatric patients. However, there are several points that should be addressed by the authors.
Major comments
Main text
- Page 3, line 73-74: “In infective endocarditis, the diagnosis is based on the modified Duke criteria, the current gold standard for diagnosing infective endocarditis”.
According to the 2015 ESC guidelines for IE: “This classification has a sensitivity of approximately 80% overall when the criteria are evaluated at the end of patient follow-up in epidemiological studies. However, the modified Duke criteria show a lower diagnostic accuracy for early diagnosis in clinical practice, especially in the case of prosthetic valve endocarditis and pacemaker or defibrillator lead IE, for which echocardiography is normal or inconclusive in up to 30% of cases. Given the recent published data, the Task Force proposes the addition of three further points in the diagnostic criteria (Table 14): (1) The identification of paravalvular lesions by cardiac CT should be considered a major criterion. (2) In the setting of the suspicion of endocarditis on a prosthetic valve, abnormal activity around the site of implantation detected by 18 F-FDG PET/CT (only if the prosthesis was implanted for >3 months) or radiolabelled leucocyte SPECT/CT should be considered a major criterion. (3) The identification of recent embolic events or infectious aneurysms by imaging only (silent events) should be considered a minor criterion.” I suggest that this section and figure 2 should be revised accordingly, because it seems inaccurate.
We have revised microbiological diagnosis section and figure 2.
- Page 5, line 146: “Although controversial, transesophageal echocardiography is valuable in children with congenital heart disease”. Why is it controversial? You mention above that in 70% of cases with IE in patients with CHD and cardiac surgery, TTE results are false-negative. I think that the whole section from line 135-152 should be rephrased, because it has poor English, and it is kind of confusing.
We have revised
- Page 6: I think that the section concerning imaging diagnosis is incomplete. The role of PET/CT in prosthetic valve endocarditis (PVE) should be explained in more details. According to the 2015 ESC guidelines for IE: “Recently, nuclear techniques, particularly 18F-FDG PET/CT, have been shown to be useful for the diagnosis of PVE. The addition of abnormal FDG uptake as a novel major criterion for PVE has thus been pointed out. An algorithm for evaluation of patients with suspected PVE, including echocardiography and PET/CT has been suggested.” (Saby L, Laas O, Habib G, Cammilleri S, Mancini J, Tessonnier L, Casalta JP, Gouriet F, Riberi A, Avierinos JF, Collart F, Mundler O, Raoult D, Thuny F. Positron emission tomography/computed tomography for diagnosis of prosthetic valve endocarditis: increased valvular 18F-fluorodeoxyglucose uptake as a novel major criterion. J Am Coll Cardiol 2013;61:2374 –2382)
Fluoroscopy is also not mentioned as a way to diagnose IE in children with mechanical valves.
We have revised imaging techniques.
- Page 7, line 206: mention the novel 2019 International CIED Infection Criteria (EHRA international consensus document on how to prevent, diagnose, and treat cardiac implantable electronic device infections, Europace (2020) 22, 515–516, doi:10.1093/europace/euz246).
We have mentioned the suggested document and added Table 1.
- Page 7, line 226-242: I suggest citing the most recent and relevant EHRA consensus document (2020) on CIED infections (EHRA international consensus document on how to prevent, diagnose, and treat cardiac implantable electronic device infections, Europace (2020) 22, 515–516, doi:10.1093/europace/euz246). You should add the recommendation for device and lead removal according to Table 8. You should also mention the duration of antibiotic treatment after the device removal (Table 9).
We have changed the document cited to the most recent suggested document (EHRA 2020 consensus document). We have added the recommendations for device removal in table 2.
- Page 7, line 226-242: What is the management in case of CIED-IE of epicardial leads? It should be added.
This document refers to infective endocarditis of endocardial leads. Pocket infection in generators with epicardial leads mandates surgical removal of both generator and epicardial lead.
- Page 9, line 281: “Aortic valve repair in children is desirable but only sometimes feasible. Replacement of the aortic valve in children is done using a mechanical prosthesis.” You should also add Ross procedure.
We have revised.
Minor comments
Abstract:
- “IE can be increased more than 140 times by congenital heart disease”. I think that it should be rephrased as follows: “The risk of IE can be increased more than…disease”.
We have revised.
- I suggest that the term “cyanotic”, instead of “cyanogenic” is used.
We have revised it.
- “The most detected organism ... Aureus”. Perhaps instead of organism, you should use the word microorganism.
We have revised it.
- “In addition, cardiac surgery and interventional cardiology associated with the placement of prostheses or conduits may increase the risk of IE up to 1.6% for Melody valve implantation.”. I think that you should incorporate this sentence with what you write in line 21-23. At the present position is kind of irrelevant.
Introduction:
- “Transposition of large vessels”. I suggest the word “great” instead of “large”.
We have revised.
- “Other risk factors for infective endocarditis may be followed in Table1”. I think what you meant is: “Other risk factors for infective endocarditis are presented in Table 1.”
We have revised accordingly.
Main text
- Page 3, line 74 “can be followed in Figure 2”. Again, I suggest that you use the phase is depicted or is presented instead.
We have revised it accordingly.
- Figure 2: correct aureu, reveling
We have revised our figure.
- Page 5, line 134: I think you meant “Imaging techniques”, instead of “imagistic diagnosis”
We have revised it.
- Page 6, line 179: according to 2021 ESC guidelines for pacing the threshold for pacing in patients with congenital heart block is <50bpm.
We have revised it.
- Page 6, line 188: I suggest that you should replace the term “handling” with “manipulation”.
We have replaced as you suggested.
- Page 8, line 272: “IE can develop within six months of undergoing cardiothoracic surgery in 272 children with CHD under three years, a risk that increases by five times in those children”: The sentence structure is confusing, I do not understand the meaning.
We have rephrased: “Patients under 3 years old who have undergone cardiac surgery to correct congenital heart disease have a 5-fold increased risk of developing infective endocarditis within 6 months.”
- Page 10, Figure 7: Please explain the abbreviations in the figure (VD-AP).
We apologize for the mistake, it’s RV-PA conduit ( right ventricle-pulmonary artery conduit).
- Page 11, Figure 8: Instead of “probabilistic”, use the term “empirical”.
We have revised it accordingly.
Conclusion
- Merge to one paragraph, as follows: “Through this overview, we would like to highlight the importance of prevention in patients at high risk of infective endocarditis, which can often be life-threatening, and the need to develop comprehensive guidelines for these patients. An up-to-date and modern management of IE and CIED-IE prostheses, conduits, or defect closure devices requires a proper and collaborative integration of diagnostic tools and therapeutic strategies. Multimodal imaging is a crucial part of diagnosis, where different techniques provide additional and unique information. Clinicians and imaging specialists should be aware of the strengths and limitations of each approach to appropriately interact with other specialists and therefore optimize patient management and improve outcomes.
We have rephrased as the reviewer has suggested us.
Comments on the Quality of English Language
Extensive editing of English language required.
An English revision has been performed.
Reviewer 3 Report
Figure 2: "Echocardiography revealing..."
line 156 "dough"?, maybe doubt.
line 323 "methicillin"
there are multiple episodes of different phraseology but overall very readable
Author Response
Figure 2: "Echocardiography revealing..."
We apologize, but we did not find the line indicated by the reviewer.
line 156 "dough"?, maybe doubt.
We have revised it accordingly.
line 323 "methicillin"
We have revised.
Comments on the Quality of English Language
there are multiple episodes of different phraseology but overall very readable
An English revision has been performed.
Reviewer 4 Report
Dardari et al. in their article "Infective endocarditis among pediatric patients with prosthetic valves and cardiac devices: A review and update of recent emerging diagnostic and management strategies," the new diagnostic and therapeutic strategies for IE in pediatric patients after cardiac procedures are presented.
Overall, the author provides a well-organized summary and update. For clarity, I would recommend adding a table with the most common pathogens and percentage of occurrence.
I thank the authors for the opportunity to review their work.
Author Response
Dardari et al. in their article "Infective endocarditis among pediatric patients with prosthetic valves and cardiac devices: A review and update of recent emerging diagnostic and management strategies," the new diagnostic and therapeutic strategies for IE in pediatric patients after cardiac procedures are presented.
Overall, the author provides a well-organized summary and update. For clarity, I would recommend adding a table with the most common pathogens and percentage of occurrence.
I thank the authors for the opportunity to review their work.
We highly appreciate your kind words, we decided not to include the table as another recent publication's main topic was the causative agents (https://www.frontiersin.org/articles/10.3389/fcell.2022.995508/full).
Reviewer 5 Report
Dardari et al reported their work named "Infective endocarditis among pediatric patients with prosthetic 2 valves and cardiac devices: A review and update of recent 3 emerging diagnostic and management strategies" and concluded "Our manuscript presents a comprehensive review of infective endocarditis associated with cardiac devices and prostheses in the pediatric population, including recent advances in diagnosis and management". I have the following comments:
- Minor language revision is essential with attention to spacing, especially on the title page
- Please spell out any abbreviation for its first time eg F-FDG PET/CT in line 86, VD-AP in Figure 7.
- Please discuss shortly other causes of endocarditis in both pediatrics and adults. You may refer to this recent paper (PMID: 36980734).
- In Figure 2: Please remove the separate letter "> c" in the orange box at the bottom.
- Were the presented patients consented or were their consents waived and why? Please share your thoughts on this.
- Figure 6: please expand in the legend on their uses.
- Figure 7: Please spell out VD-AP in the legend. Please specify the diagonal line in panel B of this figure.
Author Response
Dardari et al reported their work named "Infective endocarditis among pediatric patients with prosthetic 2 valves and cardiac devices: A review and update of recent 3 emerging diagnostic and management strategies" and concluded "Our manuscript presents a comprehensive review of infective endocarditis associated with cardiac devices and prostheses in the pediatric population, including recent advances in diagnosis and management". I have the following comments:
- Minor language revision is essential with attention to spacing, especially on the title page
An English revision has been performed.
- Please spell out any abbreviation for its first time eg F-FDG PET/CT in line 86, VD-AP in Figure 7.
We have revised it.
- Please discuss shortly other causes of endocarditis in both pediatrics and adults. You may refer to this recent paper (PMID: 36980734).
We appreciate that you took the effort to review our paper, but the manuscript you have referred is not the object of our research.
- In Figure 2: Please remove the separate letter "> c" in the orange box at the bottom.
We have revised our figure.
- Were the presented patients consented or were their consents waived and why? Please share your thoughts on this.
We have the patient's consent for using the echocardiographic images and personal pictures ( lesions, necrosis, etc).
- Figure 6: please expand in the legend on their uses.
We have expand the legend.
- Figure 7: Please spell out VD-AP in the legend. Please specify the diagonal line in panel B of this figure.
We have revised it.
Round 2
Reviewer 2 Report
Infective endocarditis among pediatric patients with prosthetic valves and cardiac devices: A review and update of recent emerging diagnostic and management strategies Rev1
The authors have addressed adequately, most of the issues raised. Some minor comments:
- It wasn’t possible for me to review the tables. Are they in the supplementary material?
- Figure 2: “revealing” instead of “reveling” and please add as we’ve already mentioned in major criteria (ESC G/l 2015 for IE, Table 14):
· Abnormal activity around the site of prosthetic valve implantation detected by 18 F-FDG PET/CT (only if the prosthesis was implanted for >3 months) or radiolabelled leukocytes SPECT/CT.
· Definite paravalvular lesions by cardiac CT
- Even though the authors have agreed in the correction of the following, I didn’t find the following in the revised manuscript:
Abstract:
o “IE can be increased more than 140 times by congenital heart disease”. I think that it should be rephrased as follows: “The risk of IE can be increased more than…disease”.
o I suggest that the term “cyanotic”, instead of “cyanogenic” is used.
o “The most detected organism ... Aureus”. Perhaps instead of organism, you should use the word microorganism.
Introduction:
o “Transposition of large vessels”. I suggest the word “great” instead of “large”.
o “Other risk factors for infective endocarditis may be followed in …”. I think what you meant is: “Other risk factors for infective endocarditis are presented in...”

Author Response
Dear reviewer,
We really appreciate that you took the time to review our paper and for the excellent recommendations that have improved our work.
We apologize for not addressing the comments on the first round.
Please find below our responses.
The authors have adequately addressed most of the issues raised. Some minor comments:
- It wasn’t possible for me to review the tables. Are they in the supplementary material?
We apologize for the mistake, there are no tables in the present manuscript, when we drafted the initial manuscript, we included some tables, but we decided to remove them before the first submission. We are sorry again for the misunderstanding.
- Figure 2: “revealing” instead of “reveling” and please add as we’ve already mentioned in major criteria (ESC G/l 2015 for IE, Table 14):
Abnormal activity around the site of prosthetic valve implantation detected by 18 F-FDG PET/CT (only if the prosthesis was implanted for >3 months) or radiolabelled leukocytes SPECT/CT.
Definite paravalvular lesions by cardiac CT
We revised the figure 2.
Even though the authors have agreed in the correction of the following, I didn’t find the following in the revised manuscript:
Abstract:
- “IE can be increased more than 140 times by congenital heart disease”. I think that it should be rephrased as follows: “The risk of IE can be increased more than…disease”.
We have revised it.
- I suggest that the term “cyanotic”, instead of “cyanogenic” is used.
We have revised it.
- “The most detected organism ... Aureus”. Perhaps instead of organism, you should use the word microorganism.
Introduction:
- “Transposition of large vessels”. I suggest the word “great” instead of “large”.
We have revised it.
- “Other risk factors for infective endocarditis may be followed in …”. I think what you meant is: “Other risk factors for infective endocarditis are presented in...”
We have revised it.